# Exploring Mental Health and Holistic Healing through the Life Stories of Indigenous Youth Who Have Experienced Homelessness

**DOI:** 10.3390/ijerph192013402

**Published:** 2022-10-17

**Authors:** Mikaela D. Gabriel, Sabina Mirza, Suzanne L. Stewart

**Affiliations:** Waakebiness Institute for Indigenous Health, Dalla Lana School of Public Health, University of Toronto, Toronto, ON M5T 3M7, Canada

**Keywords:** indigenous homelessness, indigenous youth homelessness, indigenous youth life transitions, indigenous mental health and healing, holistic healing, indigenous traditional knowledge, indigenous urban migration

## Abstract

Indigenous youth are the fastest growing population in Canada, yet are marked by profound and disproportionate personal, societal, political, and colonial barriers that predispose them to mental health challenges, employment and educational barriers, and experiences of housing insecurity and homelessness. It is only from the perspectives and experiences of Indigenous community members themselves that we can gain appropriate insights into effective supports, meaningful interventions, and accessible pathways to security. This paper will explore the mental health of Indigenous youth who are at risk of, or who have experienced, homelessness, as well as the lifelong perspectives, teachings, and guidance from Indigenous Elders and traditional knowledge keepers; their perspectives are weaved throughout, in order to provide a more effective means to addressing holistic healing and the mental health needs of Indigenous homeless youth. As educators, researchers and clinicians who have sought to understand this issue in more depth, our analysis aims to raise awareness about the complexities of Indigenous youth homelessness and push back against systemic barriers that contribute to homelessness, fail young people, and subject them to oppression. We also offer recommendations from a clinical perspective in order for clinicians, researchers and those working within communities to serve our Indigenous youth with a diverse set of methods that are tailored and ethical in their approach.

## 1. Introduction and Review of Literature

Indigenous youth are most precariously placed at the crossroads of mental health, housing, and life transitions. Canada’s bloody and buried colonial efforts ensured mass genocide of Indigenous Peoples, including personal, cultural, communal, and societal assaults that persist in a dearth of supports, opportunities, health disparities, and gaps across health, wellbeing, security, and success [1,2,3]. Centuries of colonial cultural oppression, assimilationist policies, and a legacy of injustices are still present and persistent today; such factors present multiple housing barriers for Indigenous Peoples [4]. Indigenous Peoples, which “describes three distinct cultural groups: First Nations, Métis and Inuit” ([4] p. 89) continue to experience profoundly disproportionate levels of homelessness across urban centres in Canada, but also continue to experience colonially rooted factors of houselessness (i.e., forced displacement; disconnection from ancestral territories; continued land and water defending; residential schools and forced child removal); and disruptions in sociocultural representation in urban centres (i.e., lack of representation; persistent negative stereotypes; structural violence and discrimination).

Amidst the mire of structural inadequacies of housing on reserves, forced land relocation, and current experiences of housing disparities (both personal and societal, both interactionary [5]) Indigenous Peoples have also been understood to have been “robbed of the experience of ‘home’ [3] and have been described as a “spiritual homelessness”, that is, “a crisis of personal identity wherein a person’s understanding or knowledge of how they relate to country, family, and Indigenous identity systems is confused or lacking” and significantly impact one’s mental health and wellbeing [3,6], p. 10. Therefore, although it may appear as if Indigenous youth face many of the same challenges as non-Indigenous youth, in fact, Indigenous youth experience pathways into homelessness that are inextricably compounded by the ongoing ramifications of Western colonization, persistent cultural oppression, and intergenerational trauma. Newhouse [7] described the extensive colonial efforts of eradication and forced assimilation efforts as “the Long Assault”, including residential school systems, Sixties to Millennium Scoops, forced sterilizations and mass incarceration [3,8,9]. While each subsequent effort to eradicate Indigenous Peoples is more heinous than the next, residential school system alone “accomplished what is today considered cultural genocide against Canada’s Indigenous Peoples” [9,10], p. 3.

Throughout this paper, the authors draw and adapt from the Canadian definition of youth homelessness, whereby “youth homelessness refers to the situation and experience of young people between the ages of 13 and 24 who are living independently of parents and/or caregivers, but do not have the means or ability to acquire a stable, safe or consistent residence” [11], p. 1. However, the authors expand this definition to also include those up to the age of 29 to be more inclusive of cultural considerations of youth and adulthood. This paper also integrates Thistle’s [12] definition of Indigenous homelessness:

Indigenous homelessness is not defined as lacking a structure of habitation; rather, it is more fully described and understood through a composite lens of Indigenous worldviews. These include: individuals, families and communities isolated from their relationships to land, water, place, family, kin, each other, animals, cultures, languages and identities. Importantly, Indigenous people experiencing these kinds of homelessness cannot culturally, spiritually, emotionally or physically reconnect with their Indigeneity or lost relationships.[12,13], p. 6.

Homelessness has been recognized as the total erasure of social and economic resources and supports, and as “the least desirable life circumstances one could imagine” [14] p. 89. While this is true, homelessness is also an experience that is inherently traumatic [15,16], and that Indigenous homelessness is even more multilayered and complex due to overlapping colonial and intergenerational trauma. However, some traditional Indigenous Elders and knowledge keepers promote that Indigenous homelessness is better termed as *houselessness*; despite the absence of structure, Indigenous Peoples continue to live close to our true home and first mother, the earth [6,17].

The gravity and magnitude of colonization’s racist and oppressive onslaught continues to significantly impact Indigenous youth’s health and wellbeing even today [4,9], creating major disruptions of family life and mental health as well as educational disengagement, drug misuse, and independent living [4,9,18,19]. The systemic structures that displace and harm Indigenous Peoples in securing housing is evidenced by disproportionately high rates of Indigenous homelessness across urban centres in Canada, making up as much as 35–60% in Western Canada, and 90–95% of the visibly homeless in the Northwest Territories, and 15% in Toronto [20,21,22,23]. In addition, research consistently shows that Indigenous youth are overrepresented in the homeless population in Canada and at higher risks for homelessness as colonization, ongoing racism and the lasting effects are at the core of experiences of poverty, housing precarity, and homelessness for many Indigenous youth and their families [4,9,12,19,24,25].

Indigenous homeless youth are navigating not only an oppressive and unjust history [9] p. 15, but various individual, societal, structural, and systemic barriers that force young people into homelessness. While Indigenous youth are identified as the fastest growing population in Canada [1,2], changes to traditional employment markets and occupations force mass migration of youth seeking advancement and achievement to urban centres, a trend that has been noted for decades [26,27]. On arrival, a lack of affordable housing, underemployment, discrimination, and inadequate education have become factors and failures that heavily impact the available outcomes and opportunities for young people. Discrimination in the form of racism, homophobia, and transphobia are also contributing societal and structural barriers that may not allow Indigenous youth to stay in their families or communities, and may meet them in urban settings [4,9,12,19,24]. As historical and ongoing racism, inequity and colonialism in Canada are at the root of the homelessness experience for many Indigenous youth and their families [4,28], Indigenous youth face higher mental health challenges, intergenerational, societal, and colonial impacts that are vested upon them to navigate with major barriers in accessing meaningful and representative mental health services.

Substantial literature outlines the devastating mental health impacts that accompany housing loss. While the event itself is an inherent stressor and traumatic event, it further exacerbates existing mental health challenges, of which those with adverse early life events unfairly and disproportionately experience, with “higher rates of foster care, group home placement, neglect, sexual and physical abuse, less effective kin supports, drug use, runaway behaviours, poverty, residential instability, family violence and parental psychopathologies than do psychiatric patients without histories of homelessness” [29,30,31,32], p. 372. Ongoing research has identified that homeless clients experienced various types of abuse (emotional, physical, sexual) in childhood, domestic violence, involvement in the legal system, and family members experiencing substance use concerns. Established literature reports rates of 50–90% of individuals experiencing sexual abuse among the homeless population [33,34,35,36,37]. When considering the strongest impacts to mental health and wellbeing, those who go on to experience homelessness and housing insecurity are not only some of the most stressful states of living imaginable, but a cataclysmic failure of support, personal suffering, and challenge finding further needs and services. In addition, while Canada has had a National Housing Strategy since 1999, which actively aims to support communities across the country towards addressing homelessness, this strategy does not have a strategic focus on youth homelessness since its early phases [37,38]. In terms of housing and homelessness policy action being taken in Canada, a new homelessness strategy was recently launched called Reaching Home, a community-based program aimed at preventing and reducing homelessness [37,38,39]. Despite these Canadian policy initiatives towards housing and homelessness, this review of literature and the authors’ research makes evident that the housing needs of Indigenous populations in Canada have still not been met.

Although Indigenous Peoples continue to promote cultural resurgence, challenge inequitable governmental control, and endeavour to ensure continuity of languages and cultural practices, the impact of the residential school system and racist policies continues to profoundly affect Indigenous communities [37]. It is important to emphasize that, in the Canadian context, implementing preventative responses cannot happen without holding government institutions and systems accountable to respect the human rights, leadership, and worldviews of Indigenous communities [24]. Support for Indigenous youth must also be informed by Indigenous cultures, worldviews, and relationships [12,37]. This article seeks to review ongoing factors that challenge indigenous youth mental health during housing and housing loss transitions, including perspectives and recommendations from the most important experts on Indigenous homelessness, Indigenous youth themselves. In addition, included in this paper are the lifelong perspectives, teachings and guidance from Indigenous Elders and Traditional Knowledge Keepers, whose invaluable perspectives provide insights on holistic healing and the mental health needs of Indigenous youth who are experiencing homelessness.

## 2. Theoretical Framing

The topic of Indigenous youth homelessness can be viewed through various theoretical frameworks. Gabriel’s research was rooted in two theoretical orientations: Indigenous traditional knowledges and social constructivism. Indigenous traditional knowledges are broadly defined as those perspectives, practices, and teachings passed intergenerationally through Elders and traditional knowledge keepers since time immemorial [40]. The differences between Indigenous traditional knowledges and Western societal perspectives are significant, and have caused profound harm toward Indigenous Peoples in research through harmful, extractive, and misrepresented information, as “communities were seldom consulted with and had very little, if any, control over the research process” [41]. To ethically and culturally root the research findings and perspectives in Indigenous knowledges, the inclusion of four Indigenous Elders and traditional knowledge keepers were included among participants, the teachings of which helped to shape and reflect the interpretation of this work. Alongside prioritizing and including Indigenous Elders, the understanding of health as holistic, as seen in the teachings of the medicine wheel, assists in appreciating that health is comprised of connection to the environment, planet, cosmos, and is achieved through mental, physical, emotional, and spiritual wellbeing; illness arises from disconnection or imbalance across these spheres [40].

Social constructivism is present to promote that those individual youth and Elders interviewed have unique, personal experiences, shaped by their own perceptions, language, culture, and attribution of meaning [42]. Together, these orientations offer culturally focused and theoretically flexible understandings of participant experiences, and highlight those factors they identify as most important to them; thereby we are able to amplify their voices and perspectives. Similar to Gabriel, to maintain open and flexible understandings of the research and phenomenon of Indigenous youth homelessness, Mirza’s research explored theories of social exclusion and marginalization as the primary conceptual and theoretical frameworks. Such theories can help to better understand this topic in relation to advocacy, social justice, and human rights [37]. Indigenous youth who experience homelessness face multiple and often intersecting forms of exclusion [37,43,44,45,46]. Existing and emerging literature about youth homelessness argues that the circumstance of homelessness is one of the most serious manifestations of exclusion and marginalization, complicated by various systemic, structural and individual factors that often extend beyond the lack of physical or material needs [24,37,47,48,49,50,51,52,53]. Social exclusion as a result of homelessness is a systemic process that shuts people out and bars them from society’s social, economic, political and cultural institutions [37,49,54,55,56]. With this also comes the restriction of participation in one’s community, the violation and denial of access to basic human rights, services, and institutions, and being denied dignity and respect [24,28,37,44].

Social exclusion can also be understood as stemming from extreme poverty and accounts for deprivations in other aspects of social life such as health, education, and employment [24,28,37,57]. The daily realities of young Indigenous people who experience homelessness are patterned by exclusions not only with historical overlaps, but with multiple and intersecting factors that affect employment, housing, racism, ethnicity, gender, ability, citizenships, sexism, youths’ access to spaces, institutions and practices [24,37]. For youth who participated in Gabriel [5] and Mirza’s [37] research studies, experiences of exclusion, oppression and marginalization were revealed through interviews where youth spoke to the various systemic and structural barriers faced. The diverse theoretical frameworks explored by the authors provide a deeper lens into the lived reality of Indigenous youth who are homeless and their health and mental health needs. Acknowledging and deeply understanding the injustices of homelessness requires open, fluid, and ethical theories and methodological approaches to research. In hearing the voices of youth, the researchers believe the more obscure and complex processes of exclusion and marginalization that work to powerfully create barriers for youth can be better understood. Through the multiple theoretical lenses included by authors, we can read the stories of youth about homelessness and mental health from their vantage points.

## 3. Materials and Methods

This paper aims to closely analyse the work of the co-authors Drs. Suzanne Stewart, Mikaela Gabriel, and Sabina Mirza. All of the research presented here is original research, and the data were collected by co-authors through their respective studies. Dr. Stewart is a member of the Yellowknives Dene First Nation and is strongly committed to advancing Indigenous healing through health and psychology. Dr. Mikaela Gabriel is Italian and Mi’kmaq of *Ktaqmkuk*, and her work focuses on how cultural connection and Indigenous traditional knowledge support stabilizing transitions to housing, education and employment for Indigenous Peoples. Dr. Mirza is of South Asian background, and her academic efforts focus on community-based research, youth homelessness, Indigenous youth homelessness, education and mental health. The research included in this paper was collected from various urban locations between 2017–2021. Gabriel’s research [5] explored perspectives of Indigenous homelessness, mental health, and life transitions from four Indigenous traditional Elders, knowledge keepers, and nine Indigenous youth (aged 15–29) across varying housing experiences, from absolute homelessness to couch surfing and insecure housing, consistent with broadening understandings of insecure housing and homelessness [3]. Gabriel’s research was completed during the COVID-19 pandemic, in which recruitment occurred through outreach workers, community connections, and peer recruitment to ensure COVID-19 protocols were followed for physical safety. To ensure emotional and cultural safety, Gabriel made sure to build a strong rapport with community members, using a trauma-informed, culturally focused approach, as well as active listening skills and support during proceedings. Interviews were individual and lasted between 60–90 min via Zoom, and were transcribed and analysed by a narrative coding and thematic analysis technique developed by Stewart, detailed elsewhere [58]. Results were then shared with participants for review, approval, reflection, edits, or withdrawal.

Mirza’s [37] study of youth homelessness and barriers to education included Individual surveys and in-depth interviews with 40 young people experiencing homelessness, of which 11 youth (more than 25%) identified as Indigenous. Although 40 youth were interviewed, only the narratives of those 11 Indigenous youth will be shared throughout this paper. Prior to beginning data collection, Mirza’s research with human participants had been approved by the Ethics Review Committee at York University, and throughout the duration of the project, she followed York University’s *Guidelines for Conducting Research with People Who Are Homeless.* To understand the first-hand experiences and perspectives of young homeless people, Mirza used the qualitative approach of ethnography. Mirza’s [37] study was not a classic ethnography as she did not live amid the community she did research with, but enough time was spent in the field to gain rich and detailed accounts of youth’s lives. While in the field, Mirza completed all surveys and engaged in in-depth interviews with young people directly. The quantitative survey asked about age, ethnicity, gender and sexual identity. As some groups (e.g., Indigenous and other racialized youth, LGBTQ2S+ youth) are over-represented in this population [24], the survey also asked youth if they identified as Aboriginal, Indigenous, First Nations, Inuit, or Metis. This is important; it does matter whether a youth who is homeless is Indigenous, or female versus male, or identifies as gender-fluid, as there are often multiple and intersecting layers that affect these populations, which creates substantial differences in the way youth experience homelessness, their pathways and access to resources, and the exclusions they face [37]. Using mixed methods approaches can help us better understand differences among the youth homeless population in order to find equitable solutions, especially in the context of Indigenous youth homelessness and mental health.

Participants were recruited from four shelter service agencies located in the York Region that provide youth experiencing homelessness with programming, intervention, and support. A research package was created and distributed at the shelters prior to the study outlining the goals and details of the research. The package included a corresponding flyer for the recruitment of youth participants; this allowed youth to provide shelter staff with their intention to participate. At the onset of the research, purposive and strategic sampling techniques were used, as a means of selecting groups or categories to study on the basis of their relevance to research questions, theoretical position and analysis [59,60]. Knowing there would be youth at these shelters, Mirza’s sampling also included convenience sampling, which relies on available subjects [61]. Alongside the diverse sampling that was already being implemented, youth who had completed interviews began to recruit additional participants, which is when snowball, chain and referral sampling unfolded [61]. Continued participation from youth beyond the original research recruitment was a significant indication of how widespread the problem of youth homelessness is across the region. Interviews lasted between 60–90 min and were audiotaped with participants consent, and later transcribed and analysed using narrative and thematic coding and analysis [62]. Quantitative data were entered through Survey Monkey and analysed for basic descriptive statistics to understand frequency distributions. Qualitative data were analysed thematically using Microsoft Word. Interview questions asked about experiences of homelessness, access to resources and supports, and details of family life, mental health challenges and educational barriers. Strict shelter protocols were followed when entering the site and a private and confidential room for interviewing was always designated by shelter staff. Youth were provided with $20 cash incentive to participate, and were able to withdraw participation from the research at any point.

Through their studies, Gabriel [5] and Mirza [37] were exposed to the complex health challenges of young Indigenous people from various histories and heritages. The data were collected across York Region and the Greater Toronto Area. Given the widespread problem of homelessness in Canada, it was not surprising that most youth in the study reported significant negative determinants of poor health related to social exclusion, discrimination, violence, and trauma. In fact, severe forms of pre-street adversity abound for this population, including situations of poverty, neglect, abuse, bullying, and marginalization [19,63,64,65,66,67]. After the surveys and interviews were completed, the researchers remained in contact with the staff and youth, to further volunteer and establish relationships despite the research being over at that time.

To highlight the voices and perspectives of the Indigenous people who took part in each study, direct quotes and first-person accounts which speak to the themes outlined will be presented throughout the results section. Importantly, when Indigenous youth voices are highlighted, the paper will aim to identify each youth as they identified themselves in the surveys and interviews when possible (i.e., First Nation, Cree, Inuit), to respect Indigenous identity, diversity, and naming practices [37].

## 4. Results and Analysis

The results and perspectives of participant narratives will be divided according to personal and societal domains. By differentiating broader themes related to personal and societal factors, we are better able to understand and address existing barriers and provide ethical solutions. Notably, in appreciation of Indigenous perspectives of holistic health, personal dimensions will be further divided according to the four domains of health in accordance with the Medicine Wheel: physical, emotional, spiritual, and mental health domains [40]. This will further help to illustrate the personal impacts of factors and then the compounding impacts, and various barriers.

### 4.1. Personal Domains of Health

#### 4.1.1. Physical Health

Existing research shows that the longer a young person remains homeless, the more their physical and mental health deteriorates; this puts youth at heightened risk to experience exploitation, trauma, and addictions, drop out of school, and become chronically entrenched in street life or homelessness [24]. The physical health issues of the youth in Mirza’s [37] study were extensive, and exacerbated by homelessness. Such factors were exemplified by Craig’s story, an Indigenous youth who had chronic nerve damage from a previous fall. Craig was asked what a typical day looks like, and he said: “A typical day for me is not that of a typical homeless person, I sadly do have a physical disability…I have nerve damage which makes it even harder to get on my feet and do what I need to do to get out of this situation” (Craig, age 23). His chronic pain had been impacted by chronic homelessness.

Though Craig described his situation as atypical, health deterioration and complex health detriments were common and prevalent among the homeless youth Mirza interviewed. Another Indigenous youth, Daniel, had been in a motor vehicle accident years ago, and was required to re-learn basic motor skills after major brain injuries resulting in memory loss and impairment. Daniel was required to take daily medication, which was profoundly challenged by the precarity amid the instability and precarity of homelessness. Daniel described the difficulty of maintaining medication while homeless, as dose instability can have encroaching health consequences,

I haven’t been on my medication in the longest … because I’ve been going from house to house, places like this, moved all around and my medication goes to one spot, oh he’s in a different town … we can’t give him any more meds because he has to go back to the city … they’re spending money from here to Newmarket to … 26 extra bucks for a bus ticket and then getting to Toronto and then walking how far in Toronto to get to wherever and then get my medication and then more money for a bus to get to Union Station, then from Union to wherever I will be staying and then to Newmarket and then uh Northern where I am…just in busses alone is like 75 bucks.(Daniel, age 26, Inuit youth)

For youth like Craig and Daniel, education and employment—which can aid youth in finding pathways out of homelessness—cannot be prioritized in the face of existing health concerns. Fatigue and sleep loss were also prevalent issues, contributing to vulnerability and producing negative mental and physical effects; youth described hypersomnia, insomnia, inconsistent or restless sleep rooted in the anxiety and stresses of homelessness and mental health. Interviews revealed layers of marginalization, trauma, victimization, stigmatization, poverty, poor health, chronic stress, and fatigue with gender minority and Indigenous youth impacted further due to prejudice [18,40].

Many youth who took part in Mirza’s [37] study also described being deprived of nutritious and well-balanced meals, speaking about hunger and food insecurity as a major mental health and physical health stressor. In fact, youth narratives exemplified how food insecurity and homelessness have serious human rights implications that are layered with harmful issues related to social exclusion, discrimination, power, and control [41,42,43,44,45,46]. Poor nutrition leading to negative health impacts hinders youth’s employment and education opportunities and pathways out of homelessness [43,47,48,49,50,51]. Many youth discussed their nutritional vulnerability in relation to having no choice but to eat shelter foods that were not the tastiest or healthiest; there was an ongoing anxiety and physical stress and suffering related to food deprivation.

Youth were well aware of factors regarding donations, visible expiration dates, and low-quality foods resulting in deteriorating health. Being aware of their lack of autonomous choice in food consumption, participants discussed a “beggars can’t be choosers” mentality, as if homeless people should be grateful for whatever comes their way, adding to the existing stigma, otherness, and alienation they felt. Homeless people are often blamed and dehumanized as individually responsible for their own circumstances, despite the systemic barriers that persist for this population [37]. Through participant stories, we learn that homelessness brings forth food insecurity and the lack of resources leave youth feeling helpless, and choice-less. Undoubtedly, homelessness exposes youth to violence and adversity, with harmful effects on mental health [19,37]. All of the situational and contextual barriers that homelessness brings forth are undesired and have severe impacts on the physical, emotional, spiritual, mental, and overall health and wellbeing of young Indigenous people.

#### 4.1.2. Emotional Health

This section explores the affective states, wellbeing, and impacts related to housing transitions and pathways; while in many Western settings there is little differentiation between mental and emotional health, youths’ personally reported, affective content of impacts to mood will be focused on here. Research shows that youth who are homeless experience a high degree of emotional distress [67,68]. The majority of youth in Mirza’s [37] study had faced severe forms of adversity and various social, situational, and contextual factors that had negatively impacted their mental health. Of most significant concern, 80% of the youth described feeling down, depressed, or hopeless on several days of the week. Mental health adversities such as depression, anxiety, and self-harm are often linked with difficulties regulating daily emotions [37,69], with considerable evidence suggesting that these impacts can often begin well before individuals experience homelessness [19,24]. Similarly, youth participants in Gabriel’s [5] study described compounding affective states that were not only resultant from the dire circumstances they faced in seeking stability, but also desperately needed to be addressed in order to continue on their journey to stability. One youth, an Indigenous woman in her early 20s, described the compounding sense of hopelessness and defeat in her struggle to find housing:

I was just drowning. I really felt like the whole world was against me. It just was not fair at all. I didn’t have—my family, they’re helpless to me. They’re in the same boat as me. There’s no one I could turn to. I don’t have a rich dad that could just pay my housing. What can I do to avoid becoming homeless?[5]

This youth understood that her affective state would continue to decline, and that, if left unattended, would become severe and impact her ability to navigate housing and seeking support; however, she noted that negative mood states, such as anxiety and depression, are understandably inherent to the lived reality of homeless living, “… I know if I leave it, it’s going to become a problem. I feel like there should be more interventions when it comes to people who are at risk of homelessness. That means that they’re at risk of anxiety, depression and any other type of mental illness” [5]. While navigating shelter living, this participant was newly pregnant, and had young children living with their father to avoid bringing them into shelter; she described this as compoundingly painful, as not only was she separated from her children, she was concerned about the emotional impact her anxiety, fear, and distress could have on her unborn child, as well as the risk of overwhelming hopelessness compromising her future, “sometimes you want to lose hope but then at the same time you can’t, because the moment you lose it, you’re just vulnerable to anything to drag you down” [5]. Despite her motivation for her children, she described that being homeless “was really, really hard. I wanted to just literally just give up so many times”. [5]

#### 4.1.3. Spiritual Health

Included in Gabriel’s [5] study was the important perspective of Elders and traditional knowledge keepers regarding their work supporting and healing Indigenous youth as they navigate housing transitions. In their roles as Elders and practitioners, they were brought into caring relationships with Indigenous community members navigating the complexities and overlapping health issues explored in this work; however, Elders provided crucial differences from clinical practitioners, namely in their personal involvement with clients; traditional perceptions on healing and client journeys; as well as spiritual investment and participation for ceremonies. Elders described the necessity of unconditional support for clients, “So that to me is one of the hardest things to explain, whether there’s an addiction or not, we’re going to help. And then whether you have nothing or if you have everything, you know, we’re going to help” [5]. Across all Elder participants, they promoted the use of harm reduction. For these Elders, abstinence-only services were seen as fundamentally colonial and judgmental; the ensuing judgment begets shame and rejection for community members that were struggling with mental health, substance use, and difficulty navigating services. Instead, Elders strongly promoted the crucial importance of total acceptance, non-judgment, and understanding of community members not merely as clients, but as human beings on spiritual journeys. However, other traditional practitioners prohibit participation in ceremony when others have used substances in the days prior to ceremony; these protocols become internalized for Indigenous youth, and often deterred them from participating in medicine use and ceremonies, despite their possible holistic benefit. As one Elder described:

I think one of the reasons why we’re so successful at it is because we don’t insert protocol into ceremony. I couldn’t care less if you used last night, as long as your intention of going into that lodge is pure. And sincere. And that’s all that matters. You know, there’s too much imposed protocol on our people as it is from the settlers. We shouldn’t be doing it ourselves. Who cares that you’re wearing a pair of slacks into a ceremony area. Creator doesn’t care. Why should I care?[5]

Elders described the crucial importance of traditional approaches, as they are valuable supports that extend further than clinical supports are able to. Elders hold unique, powerful roles in communities, and are still sought after when clients are distrustful of Western-based service providers; they are able to approach clients with knowledge and understanding of colonial impacts, and persist in their approach for clients who had been refused treatment, or who had negative associations with care providers due to experiences of racism. Alongside the service approaches, the interpersonal benefit of non-judgement and acceptance, made a tremendous difference for clients. One Elder described her approach to working with community members as follows:

... maybe they did abuse their partner; I don’t know. But that was yesterday; it’s not today. Who’s to say they’re going to do it today? So if I go in there with the attitude [of] if you’re a bad person because this is what you’re known to do, then they’re going to just walk out of there with a bad experience. Where we’ve really watched some of these men blossom, and they don’t know what they’re going to do when the program ends because they’ve gotten used to being accepted for who they are, being able to speak and not be judged for speaking”.[5]

Ultimately, Elders promoted an unconditional care and support for clients as essential and needed for mental health and wellbeing, as this is rooted in ongoing appreciation for human journeys and pathways with love and forgiveness:

And I think when society, especially in the city, gets back to that and funds safe spaces for people that are accepting of who they are and where they are in their journey. Because it’s your journey. It’s not my journey; it’s your journey. And you’re going to trip and fall and make the mistakes that you need to make to get you to a better place, get you where you need to be going. People need to fall and learn to pick themselves back up. But these people like all of us in the corner are supporting them.[5]

#### 4.1.4. Mental Health

This section includes aspects related to broader mental health, including cognitive domains and substance use. The authors appreciate and highlight the intrinsic connection between domains of health and wellbeing, and that each sphere of health and wellness intersects with the others. One participant in Gabriel’s [5] study, an Indigenous woman in her early 20s who had navigated homelessness, described how the experience of homelessness itself provokes mental health symptoms:

Everyone that is going through that same situation is going through the same things: depression, anxiety, any type of different mental health disorder … whatever it is, they’re going through that together… this is not something that we can necessarily control. You need to be able to control the way that you think about how we got there and why we’re here in this situation.[5]

This participant continued to describe how the perception of the world at large is impacted by continuing absence of control, supports, housing, and ongoing mental health symptoms. These compound together, and interfere with seeking housing,

If there’s some way where there’s an intervening process where you can help that person fight their battle with depression and anxiety, then they will be able to focus more on work or housing. If you have a scattered mind, or someone who’s depressed and doesn’t want to get up or do anything, you can’t expect that person to get up and do things. They’re just literally uncapable of doing that. It’s not even their fault. A lot of people are left fighting these battles alone, and then they’re left on the street and hooked up to drugs and needles because they couldn’t get out of that depression. Or that anxiety led them to think suicidal thoughts or something like that. Where do we step in?[5]

In Mirza’s [37] study, some youth were hesitant to share about their anxiety or depression, as admissions carried with them stigma and feelings of exclusion, as the combined stigma of homelessness and mental health issues can be overwhelming. However, other youths’ disclosures were more explicit, and explained that various factors can affect mood, such as physiology, access to food and meals.

#### 4.1.5. Suicidality, Self-Worth and Long-Term Effects

As described by the previous participant, unmet mental health symptoms, compounded by life stressors and housing loss, continue to worsen over time. Homeless and street-involved youth experience high rates of suicidal ideation, and suicide rates are elevated compared to the general population [24,65,70]. Research shows that rates of suicide attempts among homeless youth far exceed those of housed youth in Canada, with between 27% and 46% having attempted suicide [71,72,73,74,75], and the younger the age of the first episode of homelessness, the greater the likelihood of having attempted suicide. LGBTQ2S youth who are homeless also report a much greater degree of mental health concerns, including suicide attempts [19,24]. Approximately 25% of the youth in Mirza’s [37] study expressed having suicidal ideations on several days of the week, sharing how feelings of isolation exclusion and loneliness can lead to self-harm and suicide as their only option. Perspectives from youth in both research studies help broaden our understanding of suicide and suicidal ideation, particularly as an envisioned escape from depression and anxiety. The alarmingly high rates of anxiety and depression among this population create the conditions for youth to feel that self-harm and suicide are their last resort, with sometimes “risky behaviour” that strongly raised mortality risks and thus fit under the umbrella of suicide attempts.

Youth also spoke about suicide attempts and how homelessness leads to feeling worthless, isolated, and ambivalent about whether to continue living. While not all those who self-harm commit suicide, there seems to be a continuum of harm, aimed at escaping unbearable anxiety and depression symptoms. Further evidenced by the stories youth told, Craig shared with Mirza about his five-year depression, and about him feeling broken and worthless:

It was self-harm depression and suicidal … first like, with the depression, it wasn’t for like, it was to make myself feel pain, I would bash my head on stuff, anything, just so I wouldn’t feel emotional and then I wouldn’t start feeling that pain anymore so I just … I would just bash body parts and then it got to the point where I wouldn’t feel it, so I started cutting and at first, I was deathly afraid because I didn’t want to die when I first started doing it but I also didn’t want to feel the pain so it was challenging and then a year after I started cutting, I started attempting suicide, which seems to always fail, thank God, obviously, because I’m here. (Craig, age 23, Indigenous youth)

The multiple options and resources available to support these youth do not always connect with them at times of extremity, although these extremities were not unusual. Among the youth interviewed, many shared self-harm events, either provoked by hopelessness or intended as a cry for help. Furthermore, Elder narratives from Gabriel’s study included addressing engrained, internalized beliefs and hopelessness. One Elder described that some clients had “lost the will to live” [5] and noted these thoughts, outlook, and perceptions were a result of pervasive distrust, repeated failures in services and service workers. As explored in the next section, repeated and persistent experiences of trauma, deprivation, and social and service failures can compound and calcify into profound distrust, disinterest, and disengagement in services.

Mental health challenges inevitably bring forth long-term impacts. For instance, while youth participants had described their immediate and post-security mental health in transitions, Elders that participated in the study shared the long-term, longitudinal impacts to self-worth [5,17]. Elders described that consistent, repeated living in dwellings were “houses that nobody else wanted to live in” [5], that had no running water, or were places that others did not want to live in, impacted their self-worth by comparison, “It’s always a constant looking at the outside world through, I guess, tainted lens, that I would never be able to afford that. My family would never be able to afford to live that way” [5,17]. The lack of structural integrity, alongside the lack of belonging, compounded and wore down not only participants’ self-esteem, but also impacted their future decision making around possible housing, “I think that gets to be ingrained after a while, and then we start to accept that. And even after years of employment, and probably some really, really good jobs. But the self-esteem only allowed me to live in those second-rate or third-rate housings, because, yeah, that was—that was all we deserved. And I don’t think I did it consciously but it was subconscious. Like I said, those kinds of ways are ingrained from childhood” [5,17].

#### 4.1.6. Substance Use

Substance use and abuse were also very prevalent in the lives of young research participants. Youth who experience homelessness report challenging life histories, which puts them at exponentially higher risks in terms of exposure to violence and crime, as well as traumatic experiences (e.g., sexual and physical assaults), in comparison to housed youth [19,24,37,76,77]. Traumatic experiences often propel youth down paths of substance use and abuse, in reaction to the trauma they endured. Homeless clients may seek substance use in order to manage intolerable severity of mental health symptoms (i.e., anxiety, depression, among others) as a method of substitution for psychotropic medications, alongside the loss of support in housing [78]. Substance use and abuse might have buffered their traumatic reactions at first, but in hearing youth narratives, such encounters with substances also significantly affected their motivations and initiatives towards education, employment, and stability; it can be argued that homelessness alongside substance use can decrease motivation and make it more difficult to exit homelessness [19,37].

In Mirza’s research, the majority of youth cited using marijuana as part of their daily routine, without openly discussing other drugs they had tried or were using. As homeless youth have limited access to healthy coping mechanisms [19], using substances like marijuana, alcohol, “meth”, heroin and cocaine were some of the “tools” used to cope with mental health problems and the challenges of homelessness. Many youth used substances to cope with their circumstances as they face repeated trauma [79,80]. As told to Mirza by some youth, substance use may be the only way to warm up after being out in the freezing cold, or to sleep or kill hunger pangs [81]. Such factors are exemplified in an emotion-filled interview with James, where he told Mirza that he and his friends would “find abandoned homes” and just “stay there and get high like for days on end” to cope with the adversities of being homeless:

Ahh, it started off with weed and shrooms primarily, then uh meth, coke, which was not fun … We had guys that would come in with heroin … But you know, we did all this shit because we felt like, in a way, nobody cared because it’s like society had forgotten about us and everyone else that we had known had moved on and they don’t want to accept that this way of life exists … so it’s kind of like, that was another thing for us, was the hopelessness that developed from not really having any life skills…(James, age 24, Indigenous youth)

James’ retelling of the homelessness experiences of him and his friends reveal the importance of peer group supports for homeless youth, which are also much needed protective factors for this population. While not all youth in this study were navigating addictions, many were in different stages of dependency and/or addiction. Even so, some youth who did not initially want to do drugs were invited, tempted, experimenting and/or forced into the drug scene.

### 4.2. Social and Cultural Domains

The following factors were identified across this research, as social, cultural, communal, and societal challenges that directly influenced youth experiences of homelessness and their mental health.

#### 4.2.1. Substance Use and Cultural Barriers

While Indigenous youth reported direct experiences of substance use and their complications to mental health and accessing service delivery, cultural barriers and experiences offered unique, profound barriers with regard to acceptance, cultural connection, and support. Indigenous youth in Gabriel’s [5] study described the challenging intersections between Indigenous youth seeking cultural supports, while also navigating their substance use. Previous research has found that Elder connections in homeless outreach services can provide community connection, physical grounding, and promote crisis de-escalation in ceremonies [4,5,82,83]. Similarly, sweat lodge ceremonies have gained growing support regarding their benefits of detoxification from substance use [84,85]. However, these benefits were contested by one participant, an Indigenous youth in his late 20s, as when community members are navigating homelessness they were more focused on survival, “a lot of us aren’t exactly in the place to experience a culture when our minds are already on the street. That’s the last thing I’m thinking about is my culture when I’m already out there struggling. The culture, to me, is a thing that comes after your journey, when you’re stabilising” [5]. This youth described that homeless clients’ struggles are more focused on immediate survival, “They’re not thinking about the culture, they’re thinking about how can they meet my needs today? How can I get housed today? I want to get a house so I can attend the culture, the culture comes after you started that recovery, it doesn’t come before. A lot of people especially for the men, it hurts them to even think of the idea of culture” [5]. However, he described that many community members navigating housing loss and substance use are starkly rejected from cultural gatherings due to protocols that restrict involvement. Some traditional knowledge keepers have firm boundaries on traditional engagement and substance use; however, these protocols were described by one participant, an Indigenous youth in his late 20s, as damaging and isolative from community events, “a lot of my mindset is, why would I embrace the culture when it didn’t keep my friends, it didn’t help my family? There’s a lot of self hate within the community” [5]. The cultural rejection promoted shame, rejection, and deep pain when considering cultural services; these not only resulted in a profound sense of defeat, but an additional sense of cultural betrayal. As if Western services and society were not sufficiently racist and rejecting, cultural rejection confounded spaces of belonging and healing, greatly reducing safe spaces for community members to go.

As noted, however, Elders in this study promoted a harm-reduction approach in order to encourage cultural and ceremonial involvement. When substance use barriers around cultural and community engagement are limited, clients have increased opportunity to access Elder support, cultural services, ceremonies, and the benefit of interpersonal connection. The Indigenous cultural rejection was further compounded by societal barriers, stigma, and racism, further explored later in this paper. The Cultural Stressor Model (Figure 1) is proposed, demonstrating the compounding stressors of Indigenous homeless clients caught between cultural rejection, western service barriers, and the compounded stress of housing loss. To challenge these impacts, Elders recommended engaging in ceremonies such as Walking Them Home Ceremonies, in which clients, Elders, and community members gather to discuss disconnection from communities, pathways to legal systems and housing loss, and open discussion of community and individual needs; these ceremonies are noted to be non-judgmental, harm reductive, and profoundly rooted in open acceptance.

The Cultural Stressor Model (Figure 1) helps to illustrate the profound and compounding barriers that further exacerbate the mental health and wellbeing of Indigenous homeless clients. Elders working with homeless clients noted that multiple, compounding addictions were profound barriers, and internalized protocols, challenged Elder engagement or willingness to participate in ceremony.

#### 4.2.2. Health, Home and Education

Health adversities inevitably had a profound impact on the participants’ ability to remain engaged in, or continue education or employment. Considering the complex survival and health-related struggles homeless youth face, the majority of youth in Mirza’s study reported little interest or pleasure in doing things. Alongside the need for a bed, (e.g., to not be couch-surfing and to have more stability), many youth pointed towards wanting a place to come home to, somewhere to calm down and rest their eyes. Many youth also felt their learning disabilities hindered their success in school where individual supports and accommodations were not provided to cater to their unique needs. Consistent with findings from a national study done across Canada [24], a high percentage of homeless youth in Mirza’s study (80% of all youth interviewed) had been diagnosed with learning disabilities, in addition to Attention Deficit Disorder (ADD) and Attention Deficit Hyperactive Disorder (ADHD). While some youth were given Individual Education Plans to accommodate them better, and while schools did attempt to support individual needs, it was rarely enough to enable an effective continuation with mainstream schooling. Similar to other studies, youth felt their homelessness was not understood by educators and therefore integrating a comprehensive plan of action was not made possible [86].

Denisha, an Indigenous youth, explained that hunger and having no time and money are some of the main barriers to her being able to stay in school, despite having started a college program of her choice. In addition, she stated: “I knew I had ADD, depression and anxiety, but I didn’t get diagnosed with this until I was in the hospital being treated last year. I’ve been in the hospital five times in the last two and a half years”. Not being able to focus on what the teacher was saying and what the lesson was about was difficult enough, due to hunger and fatigue, but learning disabilities compounded the lack of focus further. Similarly, Steven said: “I’m surprised I actually passed because I have a learning disability to the point where … say I’m in class and … the teacher asks you a question … say that I’m looking that way, the teacher’s still talking to me and I still, I still can’t focus” (Steven, age 22, bisexual, Indigenous youth). Some youth were medicated for learning disabilities, with assorted sedatives; they also struggled to navigate psychophysical issues such as anxiety, trauma, and depression and more complicated conditions like learning disabilities, ADD, ADHD, developmental delays, cerebral palsy, psychosis, schizophrenia, bi-polar disorders, borderline personality disorder, or multiple concurrent disorders [18,37].

For several youth in this study, along with neglect, trauma, family adversities, and not eating or sleeping properly due to homelessness, school and being able to find employment was a lesser priority than the complex health issues they faced. Undoubtedly, homeless youth confront a range of issues that could overwhelm any experience of well-being [19,24,86]. A few of the youth also discussed their post-traumatic stress, attributed to the extreme abuse they faced in their home environment and in foster care. Many Indigenous youth pointed to various reasons for homelessness including family poverty, abuse and interpersonal violence, housing instability, and mental health and addiction. In addition, some youth were also caregivers for their parents due to the parents’ mental health disorders, which prevented youth from fully focusing on school and hindered healthy transitions to adulthood, and potential for pathways out of homelessness. These are just some of the complexities youth shared, in terms of their homelessness experiences and how that impacted their physical and mental health, resulting in diminished educational aspirations.

Youth faced overlapping and varied stresses, addictions, and mental and physical health detriments. Although many youths’ stories display resilience, that does not detract from the fact that it is difficult for homeless youth to navigate a stigmatized, low-status social identity [66]. This is clear in the helplessness and despair the youth expressed, and in how health detriments led to homelessness, and then reflexively homelessness further denigrated their physical and mental health. Feelings related to being misunderstood, loneliness, alienation, isolation, and despair, alongside the loss of family and community, further compounded the health issues faced. These issues can translate into greater and long-lasting mental health challenges and harms, especially as health service systems and providers in most settings are poorly equipped and resourced to meet their mental and physical health needs [19]. The interviews reaffirmed that Indigenous youth who are forced to endure unstable living conditions, experience homeless and street-involvement faced disproportionately higher rates of multiple health-related challenges, when compared to their housed peers [19,37,67,87,88]. In addition, due to youths’ vulnerability, their prospects of healing are bleak.

#### 4.2.3. Services

Participants in Gabriel’s [5] study described a failure of the social safety net, and a lack of preventative services in place to prevent homelessness for clients. One Indigenous youth described,

It was very hard trying to get the necessary help that I needed, because I didn’t know where to start. I had no resources; I had no help; I had no workers; I had nothing. No one. I had to really do a lot of independent research on my own, even though I wasn’t in shelter. They’re supposed to be doing all the helping and they’re supposed to, get me back on track and to avoid homelessness—because I’ve never been homeless before—to intervene before it’s something that continues to happen. Because I’m not stable; I don’t have the things that I should have at my age. So, if I wouldn’t have really took it upon myself to get the right help that I needed, I probably would have ended up staying in the shelter longer than I should have.[5] p. 457.

She described attempting to navigate housing through a housing worker while also seeking to care for her children. However, she reported that her housing worker offered limited help and hope,

All she kept saying is, ‘housing is a 14-years-long waiting list, I can’t see you getting housing or your children back.’ She’s supposed to be someone who’s supporting families? No. That was really wrong and I actually, I told her about herself. I told her, ‘you automatically judged me. Your job is supposed to be non-biased. You’re not supposed to just stand there and judge people because of a little picture of what you have’.[5] p. 467.

With unsupportive workers, clients are left to advocate and seek services on their own, “especially if you don’t have good advocacy skills for yourself … You’re really depending on that one person to provide you correct intel” [5] p. 469. In the absence of appropriate structures, individual motivation becomes the sole force in seeking services, which is equally as influenced from the impacts of mental health symptoms. While the weight of individual experiences can be cataclysmic, an Indigenous youth participant described that these occur in the context of societal barriers,

It’s not because of us; it’s this economy and the way that everything is and how rigged this system is. It’s not fair for a lot of people, especially the people that are living in poverty. It just never ends well for us. So we really need somebody to give these women a chance to just become successful. Don’t we all have that right? I think about that all the time.[5] p. 471.

These same barriers existed when seeking mental health treatment to address the impacts of such societal barriers. Participants in Gabriel’s study described that structural barriers, long waitlists, and the need for referrals for mental health treatment and intervention (i.e., trauma treatment, grief support, or substance abuse support) continued to compound the immediate needs for accessing crisis services. Without accessible services, and the imposing challenges of an unfair and uncompassionate society, clients are left with minimal hope with no end of barriers to finding stability.

#### 4.2.4. Racism

Learning more about the daily realities of Indigenous youth who have experienced homelessness allows us to make sense of the ways in which history, poverty and racism interact to restrict access to the available opportunities for youth [44]. Racism, either personal or structural, continues to impact Indigenous Peoples, and has been well documented across the homelessness literature as an old, persistent problem, “Yes, probably there is discrimination [in the housing market]. We see lots of Native discrimination in hospital services. The Native homeless do not receive the same level of care as us” [89] p. 49. Participants described the importance of having Indigenous service providers to better understand the Indigenous experience,

I don’t feel like they should have workers that are not Native because most of the referrals that they get are Native parents. How can you send someone who won’t understand their point of view? I automatically become biased, because they don’t understand what that person has gone through to get where they are, what happened for them to be where they are. … We get discriminated on a daily basis. I’m not only Native but I’m a Black woman too. So, when do we get a break?[5] p. 468.

As noted in the context of Elder connection, cultural understanding, perspectives on increasing Indigenous-specific service provision, or culturally integrated, safe models of care, are crucial for reaching and supporting Indigenous youth. Personal connection and culturally safe understanding of Indigenous youth experiences are powerful supports that can offer meaningful changes in challenging homelessness pathways and cycles. However, the absence of cultural understanding and culturally specific approaches are in no way benign; indeed, they are typically cataclysmic to Indigenous mental health, rooted in stereotypes, negative perceptions, discrimination, and negative service experiences. Indigenous Peoples report negative racial experiences limiting their access to clinical care, housing and landlord connections, and advancement in employment and education domains, all of which are necessary for Indigenous domains of stability [5,17,82,88,90,91,92,93]. However, Elders in this study described the broader concerns that racism has on Indigenous Peoples; not only in direct service interruptions and lack of engaged and culturally safe care, but Indigenous dehumanization. One Elder described:

…there was a guy in a wheelchair that was sitting like this for quite a while. How hard was it for me to tap him on the shoulder and say, dude, are you OK? And he acknowledged us and off we went. He wasn’t asking for anything; he just fell asleep. What if he was dead there? People don’t give a s---. But if it was a dog or a squirrel or a raccoon lying on the side of the street, people would be all over it.[5]

The successful otherization of Indigenous Peoples, an enduring impact of societal and structural racism, continues to present itself in existing stereotypes, further compounding social understanding of the complex colonial factors that contribute to Indigenous homelessness, as well as evident by how Indigenous stereotypes and people are treated and perceived, “Indigenous and black people are singled out because of their race, and it should never be. You know? ‘Look at those drunken Indians on the street.’ How many times have we heard that one? ‘As drugged-up Indians.’ They’re panhandling or whatever. But they never stop to think that they could be in the same position” [5]. Exemplified by the words of participants in Gabriel’s and Mirza’s research studies, it becomes evident the multiple and intersectional ways by which Indigenous people are marginalized due to racism, all of which are daily realities for youth to navigate.

## 5. Discussion and Recommendations

As researchers, listening to the life stories of young Indigenous people experiencing the unfortunate circumstances of homelessness was a deeply profound experience. Each story brought forth a unique perspective into the lived realities, stressors, and barriers that Indigenous youth face in their journeys; while common themes did arise, there was also diversity in the factors that precipitated and forced youth into street living and housing insecurity. With such overwhelming personal, social, societal, and colonial factors, the authors will now attempt to frame these issues in terms of how they might inform or influence direct action, and address pathways forward as well as future research [93]. Clinical recommendations will also be provided herein.

Homelessness continues to be a major concern for many Indigenous youth, and the impacts on their mental health and wellbeing are dire. Homelessness and survival in street living poses multiple risks, culminating in shorter life expectancies [94]. Research from Gabriel [5] and Mirza’s [37] studies makes evident that the number of youth in shelters confirmed the continuous struggle to meet their needs, and persistent failures of social structures in supporting their urban transitions. Without a diverse approach that centres the impact of colonization on Indigenous youth, as well as integrates the knowledge and expertise of Elders and traditional knowledge keepers, the high rates of Indigenous youth homelessness will continue to persist, and existing services will continue to be ineffective. We must approach our response to homelessness differently; as stated, Indigenous homelessness has unique, foundational factors that grossly predispose Indigenous youth for tremendous struggle. A holistic focus on life transitions, including promoting a positive sense of culture and identity, is crucial to fulfilling the tremendous promise and honour the great gifts that Indigenous youth bring with them to urban centres.

Many youth that participated in these research efforts described the mental health challenges that came from street and shelter living. Research makes evident that the homelessness sector is not structured to support youth mental health. The common response to youth homelessness in Canada is one that includes responses to crisis, as opposed to prevention, which contributes to overwhelmed drop-in and shelter services and reactive measures taken to focus solely on basic necessities and the other significant aspects of young people’s lives (access to education and mental health promotion) are largely ignored [55,86]. Due to structural and systemic barriers in reserve and small communities, the ongoing housing crisis on reserves, and economic impact of colonization, urban Indigenous youth were often unable to depend on family members for support, placing them in a precarious position of earning an income in order to obtain basic necessities to stay alive and stay sheltered.

Although basic necessities are provided when youth enter the shelter system, attempting to meet the equally important aspects of young people’s lives, health, and wellbeing is drowned out by the necessity of survival, and falls from focus. Youth described many complex reasons that contributed to their increased risk of homelessness and mental health challenges (abusive home environments, CAS involvement, mental health issues, substance use, poverty and precarious employment), which often began a process of compounding stigmatization and exclusion. Analysis of findings revealed that there are many social, structural, institutional, colonial, and systemic barriers that force youth into homelessness, and there is a high correlation between homelessness and mental health challenges, which is further impacted by restricted access to housing, education, employment, and important community and social supports [55]. Appreciating that Indigenous homelessness is far from personal, but is instead an intersection of housing, race, ethnicity, culture, diversity, social class, gender, sexuality, accessibility, education, and families, and how all of these factors differently impact the experience of homelessness and of mental health challenges, shifts the focus of Indigenous homelessness from a projected personal view, but evidence of the profound national, systemic failure of the social safety net, service provision, and human rights. Indigenous Peoples are vastly overrepresented among homeless populations across Canadian urban centres, and Indigenous youth are no exception to these heartbreaking statistics; their experiences are the result of early, compounded by stigma, racism, sexism, and homophobia within families, schools, and foster or group homes. Such intersecting forms of marginalization, discrimination, and oppression add additional burdens, and exclusions, impacting a youth’s pathways out of homelessness and their access to education, employment and health care prior to further housing and homelessness risks.

Without housing, home, and individualized accommodations, youth are deprived of not only basic necessities, but the structures, supports, and stability with which to survive and live a life of achievement, security, and self-advancement; this puts them at risk of irreversible and compounding debt, persistent poverty, becoming entrenched in street life, and at even greater risk for violence and exploitation—evidenced by so many of the stories youth shared about their extended and chronic homelessness. Youth also described use of various substances to cope with the pain of their lived realities, the subsequent suffering from serious mental health issues, and resultant suicidal ideations and multiple suicide attempts. The authors strongly support further research and clinical interventions to address and create improved mental health promotion and outreach supports for homeless and housing insecure youth. Educators, clinicians, and researchers are recommended to see the bravery of participating youth in sharing their deepest pain and suffering held in these narratives, and strive to dismantle institutional barriers that created such suffering in the first place.

Youth remind us of the importance of reconsidering and restructuring institutional barriers and policy reform. Their words teach us about the hidden stories of resilience, courage and survival in the face of systems and structures that are ignorant to their existence, and the risks required to seek stability. Despite navigating multiple barriers that persistently ignore their needs, and personal impacts of resounding hopelessness, youth continued to display remarkable resources and hope; these gifts must be fostered by attending to youth voices, recommendations, and supporting holistic supports, not merely survival efforts. These can help to combat the risks related to health exploitation, and promote holistic healing from the trauma of street living—addressed through prevention and early interventions. Youth in these studies strongly promoted the importance of accessible and affordable housing; survival and stability in preventing homelessness was nearly impossible with ever-rising and competing market rent units. Increasing subsidized housing buildings and rental opportunities can better support youth stabilization and living on migration to urban centres. Other efforts, such as Housing First initiatives, that provide housing and connected supports for homeless individuals can seek to address providing shelter and housing for existing homeless populations. Similarly, barrier-free, community-hosted, holistic services that teach skills such as budgeting and debt management, as well as career-track opportunities, can be life changing to youth in their life trajectories [5,17].

In terms of working meaningfully with youth, Indigenous clinicians, researchers, and educators have strongly promoted the importance of cultural safety for working with Indigenous populations. Existing Western-based programs that fail to integrate cultural practices, Elder involvement, and community approaches have, and will continue, to fail in their efforts at closing the gap of Indigenous homelessness and health inequities. Elders in Gabriel’s and Stewart’s studies [4,58,95,96] recommend an acceptance-based, harm-reduction, and trauma-informed approach, that places Indigenous traditional practices and ceremonies at the centre of care and housing for Indigenous clients and community members, with Western skills and structures offering and contributing unique strengths. In terms of supporting Indigenous youth, supporting their transitions should include promotion of cultural identity and community connection; cultural safety and anti-racism initiatives in service provision; and fostering “self-esteem building, control of their lives, not to be victimized. They need positive home environments and self-esteem to help them pursue their dreams” [88] p. 50. Oelke et al. [96] described how the competing ignorance of culture, colonization, racism, and realities of Canada’s history continue to perpetuate service barriers, “lack of information of important issues (e.g., history of intergenerational trauma) and of Indigenous cultural backgrounds (e.g., social and cultural norms, protocols and expectations) were commonly identified by participants … [and] perceived and/or real racism impacted working together” [96] p. 6. These authors promote that it is far past time for mass, multilevel, mandatory education on Canada’s colonial history in educational domains; the continued ignorance of the realities of Indigenous Peoples and colonial impacts to healthcare proves to be not only dangerous if unchallenged, and blind to the suffering of thousands of peoples, but ultimately lethal.

While the multitude of issues that contribute to youth homelessness feels challenging, this is all the more reason to explore dimensions of prevention, early intervention, and outreach support through the expertise of youth from these studies. To further the conversation on prevention Gaetz et al. [55] p. 21 use the public health model to illustrate how we can think about homelessness prevention for youth, highlighting the importance of minimizing harm to individuals and communities through lowering risk of disease, illness, and injury, and identifying risk and protective factors. It is important for educators, researchers and clinicians to consider such models as they demonstrate why we must intervene earlier than we are, because intervening later evidently has dire consequences for young people, which we want to minimize and eliminate [85,97], intervening earlier could prevent the occurrence of homelessness—which is a main component of youth homelessness prevention. Also, any preventative intervention must also be housing-led and have immediate access to housing as part of the response [55] p. 22.

There are some limitations of the research that the authors address as areas for key considerations for future research. For instance, because Indigenous youth homelessness is so widespread across Canada, capturing a complete picture can be difficult. At the same time, as qualitative researchers, we always respect the value that even one life story brings to a subject being explored and therefore, we believe the perspectives of the youth in this study are valuable insights into the lives of youth experiencing homelessness and the many systemic barriers they face. As sample sizes for research studies done in specific urban areas can be small, we may also underestimate the magnitude of the problem. It is also difficult to gather highly detailed accounts of youth’s lives when engaging in 1 h long interviews as such a limited timeframe can never capture a whole picture of the struggles and circumstances of homelessness which are daily realities for our Indigenous youth. Research makes clear that we desperately need different service providers and public systems need to work in order to keep young from becoming homeless [55]. The authors hope that in Canada, we can continue working towards preventative approaches to Indigenous youth homelessness that support young people at risk of homelessness and housing precarity.

## 6. Conclusions

Indigenous youth should have beautiful futures ahead of them; futures focused on promoting cultural representation, healthy holistic growth and development, and carrying on traditional practices into the modern world for the next seven generations. Instead, Indigenous youth are faced with tremendous barriers and challenges that utterly restrict their very survival at every conceivable turn. While Indigenous youth are the fastest growing population in Canada [1,2], they face some of the most complex and devastating prospects of what that future will look like, across domains of housing, employment, education, health and wellbeing. It comes as little surprise that the mental health of Indigenous youth is impacted by the current circumstances and stressors that they are encouraged to venture into a society that continually refuses to acknowledge genocide of their ancestors, to say nothing of the ongoing colonial project and its continued, current impact. Indigenous youth homelessness and mental health must be contextualized within the appropriate historical, systemic, and societal markers: Western colonization and its pervasive effects. Ongoing research promotes that while it is necessary to house and home Indigenous Peoples, the roots of Indigenous homelessness are inextricably connected with Canada’s genocide and its persistent, enduring effects.

Racism and discrimination aimed at Indigenous peoples are firmly entrenched in Canadian society, producing impenetrable systemic and societal barriers, such as a lack of affordable and appropriate housing, insufficient and culturally inappropriate health and education services, irrelevant and inadequate employment opportunities, and a crumbling infrastructure in First Nations, Inuit, and Métis communities. The fiduciary abandonment of Indigenous communities by the state, which has greatly contributed to Indigenous homelessness, is manifested by chronic underfunding by the federal, provincial and territorial governments of Canada [12] p. 7.

When exploring Indigenous youth homelessness and mental health promotion and care, we must always remember that these young people’s lived realities and experiences been shaped by an oppressive and unjust history that is marked by colonization, intergenerational trauma and its lasting impacts. Across Canada, the ongoing colonial project continues to impact Indigenous youth and their families today and therefore we need tailored and ethical approaches to help prevent homelessness occurrences for our Indigenous youth. These authors have dedicated their lives and work to hearing and sharing the stories of Indigenous youth and young adults who experience homelessness. The life stories of these brave participants make evident the tremendous impact of multisystemic dimensions and blockades that interrupted pathways, careers, and futures of untold potential. There is no way to tell what contributions to society these youth could make, if they only had the stability to succeed; it is our obligation, then, to dismantle the structural racism that contributed to these challenges, and work with youth to create a future where those roads are a little clearer. It is incumbent for educators, policy makers, workers, and citizens of Canada to participate in the restoration and revitalization of Indigenous health, through means of education, challenging engrained racism, and reforming service delivery. Youth shared their resourcefulness, determination, and sheer endurance in the face of dire circumstances and compounding hopelessness; a final recommendation is to contribute to a society in which youth need not learn what flowers grow from the soil of such hardship. After all, we would not want our children to suffer so gravely. Let us help All Our Relations.

## Figures and Tables

**Figure 1 ijerph-19-13402-f001:**
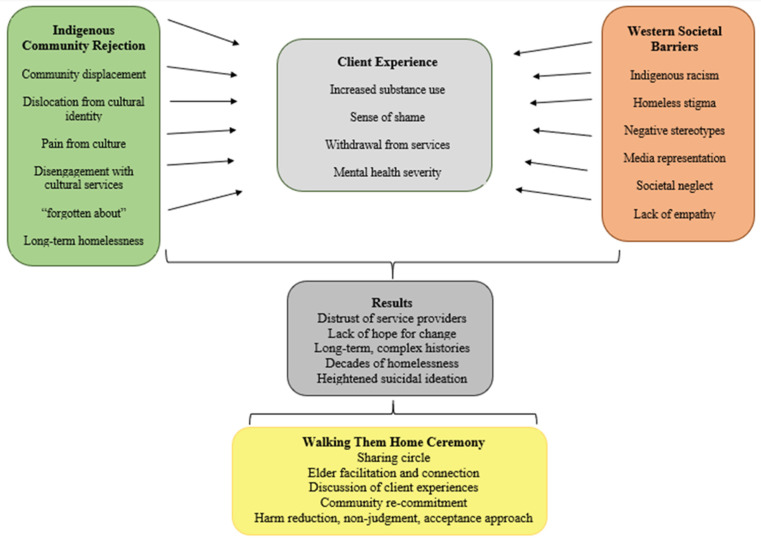
Cultural Stressor Model, in [5], employed to illustrate and describe the unique tensions experienced by Indigenous homeless youth. The enforcement of abstinence-only protocols around ceremony use, according to participants, promoted heightened sense of shame and rejection from cultural participation. Elders in this study promoted the importance of acceptance and non-judgment, including harm-reduction inclusion in ceremonies in order to support cultural connection and reunion through ceremonies such as the Walking Them Home ceremony. Reprinted with permission from [5]. 2021, Gabriel.

## Data Availability

Data sharing not applicable. No new data were created or analysed in this study. Data sharing is not applicable to this article.

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
