# Peer review of "Exploring Mental Health and Holistic Healing through the Life Stories of Indigenous Youth Who Have Experienced Homelessness"

_ijerph, 2022, doi:10.3390/ijerph192013402_

Round 1

Reviewer 1 Report

This is an excellent paper. All that is needed is information about methodology:

1. Need to add ages of participants

2. how were participants recruited?

3. Give information about the interviews. Where were they done, who did them, how long were they on average, what questions did you ask?

4. Give your definition of homelessness for this sample.

5. Did you use qualitative software for the analysis? Which one?

6. What quantitative measures did you use? Add this to the resultsw.

Author Response

Response to Reviewer 1:

Thank you for your review, we have provided additional information about methodology that can be found below.

  • Need to add ages of participants

Authors have now added in the age of participants after their quotes.

  • how were participants recruited? 

Information about participant recruitment has now been added.

  • Give information about the interviews. Where were they done, who did them, how long were they on average, what questions did you ask?

Additional information about the interviews, including where they were done, who did them, how long they were on average and what questions were asked has now been included.

  • Give your definition of homelessness for this sample.

A definition of youth homelessness has now been included to reflect the sample.

  • Did you use qualitative software for the analysis? Which one?

Quantitative and qualitative software information regarding analysis has now been added.

  • What quantitative measures did you use? Add this to the results.

Information about quantitative measure has been included in the methods section.

Reviewer 2 Report

This is an interesting article presenting very valuable research into the lived experience of Indigenous homeless youth in Canada – an important, under-researched subject. The evidence contained in the article is extensive and includes novel findings with critical implications for housing policy and practice. There is a clear need for this research, however much more work is required to bring this article to a publishable standard. Please find detailed feedback below.

1.     Intro/Background/ Structure

The research needs theorising. Many of the arguments presented here hinge on the connection between (past/current) colonial violence and the contemporary housing needs of Indigenous People, however the article lacks a conceptualisation of power. Given the focus on the real consequences of colonialism for subjugated populations, literature concerned with a ‘materialised’ postcolonial theory might be useful here to understand hegemonic forces underpinning marginalisation. See references below which might be useful.

In general, there is a lot of assumed knowledge on behalf of the reader with respect to the policy and practice landscape in which this research is situated, as well as the cultural practices of the Indigenous People involved in the study. For example, it would be useful if there was background provided on Canadian homelessness policy and a discussion of the housing needs of Indigenous populations. Having this contextual information would greatly help the reader to understand the significance of Figure 1 – which also requires an explanation of the Walking Them Home Ceremony (and other Indigenous practices concerned with Holistic Healing).

The article would also benefit from a restructure. The hierarchy of subtitles makes the presentation of the article appear more like a report than an academic piece – 4 tiers is far too many. Additionally, the discussion under the subheadings at times does not correspond with the titles. For example, the two highest level subheadings (‘personal domains of health’ and ‘social and cultural domains’) seem to broadly discuss the impact of homelessness on youth and factors which influence their experiences. Much of the discussion in Section 3 (results and analysis) appear to go far beyond the scope of the title, which suggests an emphasis on mental health. It’s not clear, for example, how education is relevant here. The scope of the article seems to have been expanded due to a tendency to summarise the entire work of Gabriel and Mirza, which, as I understand from the article presented here, is concerned with more than the mental health consequences of homelessness.

2.     References/citation

The presentation of references and in-text citation must be improved. The reference list seems to be a compilation of endnotes but the numbering does not correspond with the order in which the references appear in text. Additionally, the in-text citation is confusing – for example, multiple sources seem to be referenced but only one set of page numbers in the reference is included (see for example page 1 line 41). The references also sometimes include a dash rather than a comma making it appear like a page number range (see page 10 line 471). This style of referencing also makes it unclear when primary or secondary source data is being referenced. Respondent quotes are sometimes followed by attributional data that one would expect to see in the analysis of qualitative data (for example, p. 5 line 231) but most of the time is followed by a number (which seems to correspond with an endnote) and a page number.

3.     Methods/Research design

Related to the question above about attributional data, the methods need to be clearer about the data presented in the research. Has original data been collected for this article? If so, more information is needed on the sample size, demographic information about participants, recruitment, data collection and so on. The way in which participant quotes have been referenced here (referring to a page number within a publication) seems to suggest that most of the data presented is secondary – quotes taken directly from already published material. Much of the discussion within the methods section therefore seems to be about other studies rather than the one at hand. Or is this a fresh analysis of existing data that has been collected for other purposes? If this is the case, then original source data needs to be included in the analysis (rather than taken from another publication) and attributional data presented in the same way (i.e. p. 5 line 231). Relatedly, attention also needs to be given to the quotes included in the article – at times the evidence provided doesn’t seem to be the best fit for the point being made in the narrative (for example, p.6 line 302 seems more about the importance of social support networks and less about hopelessness). Editing long quotes would help tighten the evidence presented to support arguments (see p.7 line 352). The title makes reference to life stories as a method, however there is no discussion of this approach within the methods section, and only appears later on page 18 (line 863).

4.     Results

There is a lot of information included in this section and important findings seem to get lost in the detail. Restructuring will help to refine focus (see comments above). After reading the article it was still unclear to me how the experiences of homelessness and the housing need of Indigenous youth differs from that of other populations – which seems central to the argument. Later in the section there also appears to be quotes from Indigenous Elders, which was surprising given the methods seem to suggest qualitative interviews were only conducted with youth experiencing homelessness (which also seem to include people who were not Indigenous? P. 4 line 164). The data presented here seems critical to the second part of the title relating to ‘holistic healing’. It was very interesting to read about these interventions from the perspective of Elders (for example p.11 line 552). It would be useful to have more data presented here from the youth themselves on how they perceived interventions involving ‘holistic healing’ – there seems to be a short discussion here (starting p.11 line 523) but the quotes provided here all seem to revolve around negative experiences. If the article is promoting the use holistic healing practices as a more effective means to address the mental health needs of Indigenous homeless youth, then I would have expected more positive experiences being presented as evidence for that conclusion.

5.     Conclusions

Section 5 (conclusion) needs to connect specific findings (articulated within a theoretical/conceptual framework) to specific recommendations (for policy, practice, research etc). As currently presented, section 4 (discussion and recommendations) seems to include elements of the conclusion. The statement on page 15 (line 738) that ‘it can be difficult to know how best respond’ to the problem suggests that more thinking is required here to fully set out implications for policy or research agendas concerned with Indigenous homeless youth, in Canada or elsewhere.

6.     Editing/language

Some edits are required for clarity for example:

·      Point made starting p. 1 line 35 is unclear

·      What is meant by spiritual homelessness? (p. 2 line 45)

·      What are ‘Millennial Scoops’? (p. 2 line 54)

·      Discussion refers to ‘barriers’ throughout – barriers to what? To accessing housing? To improved mental health?

·      Section 2 (methods) flips between using first and second names of authors, which is confusing for the reader.

·      What is the ‘Medicine Wheel’? (p. 4 line 192)

·      80% of how many? (p.5 line 213)

·      Casual language used in places, for example ‘try to imagine’ (p. 5 line 221)

·      What are ‘internalised protocols’? (p. 7 line 339)

·      What is meant by ‘post-security’? (p. 9 line 452)

·      Clients of what? Services? See comment above about needing more background information.

In conclusion, this is really important work that I really hope is published. The article I believe includes seldom-heard voices that need to be heard by policy makers, service providers and researchers. And potentially could make a very valuable contribution to a materialised critique of postcolonial theory. I very much look forward to seeing how this work develops.

Potentially useful sources:

Blunt, A., & McEwan, C. (2002). Introducing postcolonial geographies. In A. Blunt & C. McEwan (Eds.), Postcolonial Geographies (pp. 1–6). Continuum.

Cook, I., & Harrison, M. (2003). Cross over food: re-materializing postcolonial geographies. Transactions of the Institute of British Geographers, 28(3), 296–317.

Hall, S. (1996). What was ‘the post-colonial’? Thinking at the limit. In I. Chambers & L. Curti (Eds.), The Post-Colonial Question: Common Skies, Divided Horizons (242–60). Routledge.

Jacobs, J. M. (1996). Edge of Empire: Postcolonialism and the City. Routledge.

Kennedy, M. (2015). Urban poverty and homelessness in the international postcolonial world. In Reworking Postcolonialism (pp. 57-71). Palgrave Macmillan.

McEwan, C. (2002). Postcolonialism. In V. Desai and R. B. Potter (Eds), The Companion to Development Studies (pp. 127–131). Arnold.

McEwan, C. (2003). Material geographies and postcolonialism. Singapore Journal of Tropical Geography, 24(3), 340–355.

San Juan, E. (1998). Beyond Postcolonial Theory. London.

Author Response

Thank you for your detailed review of our article and your comments. We have included some responses to your review below and have tried our best to revise and edit to include your suggestions throughout the article, which are reflected in the edited submission.

  1. Intro/Background/ Structure

The research needs theorising. Many of the arguments presented here hinge on the connection between (past/current) colonial violence and the contemporary housing needs of Indigenous People, however the article lacks a conceptualisation of power. Given the focus on the real consequences of colonialism for subjugated populations, literature concerned with a ‘materialised’ postcolonial theory might be useful here to understand hegemonic forces underpinning marginalisation. See references below which might be useful.

We, the authors, have now included the theoretical framing from our respective research studies. Gabriel’s includes Indigenous Traditional Knowledges and Social Constructivism and Mirza’s includes a discussion on theories of social exclusion and marginalization. We believe these theories to be significant in being able to conceptualize power in the context of homelessness as it related to young people who are Indigenous and their housing needs as they navigate homelessness. 

In general, there is a lot of assumed knowledge on behalf of the reader with respect to the policy and practice landscape in which this research is situated, as well as the cultural practices of the Indigenous People involved in the study. For example, it would be useful if there was background provided on Canadian homelessness policy and a discussion of the housing needs of Indigenous populations. Having this contextual information would greatly help the reader to understand the significance of Figure 1 – which also requires an explanation of the Walking Them Home Ceremony (and other Indigenous practices concerned with Holistic Healing).

The article would also benefit from a restructure. The hierarchy of subtitles makes the presentation of the article appear more like a report than an academic piece – 4 tiers is far too many. Additionally, the discussion under the subheadings at times does not correspond with the titles. For example, the two highest level subheadings (‘personal domains of health’ and ‘social and cultural domains’) seem to broadly discuss the impact of homelessness on youth and factors which influence their experiences. Much of the discussion in Section 3 (results and analysis) appear to go far beyond the scope of the title, which suggests an emphasis on mental health. It’s not clear, for example, how education is relevant here. The scope of the article seems to have been expanded due to a tendency to summarise the entire work of Gabriel and Mirza, which, as I understand from the article presented here, is concerned with more than the mental health consequences of homelessness.

The authors have revised some aspects of the article that discuss education and have placed a stronger emphasis on mental health. 

If the reviewer could suggest or recommend a way in which we could restructure the article for clarity and readability, we would greatly appreciate the feedback so that we may incorporate the suggestions in our final version.

  1. References/citation

The presentation of references and in-text citation must be improved. The reference list seems to be a compilation of endnotes but the numbering does not correspond with the order in which the references appear in text. Additionally, the in-text citation is confusing – for example, multiple sources seem to be referenced but only one set of page numbers in the reference is included (see for example page 1 line 41). The references also sometimes include a dash rather than a comma making it appear like a page number range (see page 10 line 471). This style of referencing also makes it unclear when primary or secondary source data is being referenced. Respondent quotes are sometimes followed by attributional data that one would expect to see in the analysis of qualitative data (for example, p. 5 line 231) but most of the time is followed by a number (which seems to correspond with an endnote) and a page number.

The citations have all been edited to reflect proper format for the journal, and they should now correspond with the way the references appear throughout the article.

  1. Methods/Research design

Related to the question above about attributional data, the methods need to be clearer about the data presented in the research. Has original data been collected for this article? If so, more information is needed on the sample size, demographic information about participants, recruitment, data collection and so on. The way in which participant quotes have been referenced here (referring to a page number within a publication) seems to suggest that most of the data presented is secondary – quotes taken directly from already published material. Much of the discussion within the methods section therefore seems to be about other studies rather than the one at hand. Or is this a fresh analysis of existing data that has been collected for other purposes? If this is the case, then original source data needs to be included in the analysis (rather than taken from another publication) and attributional data presented in the same way (i.e. p. 5 line 231). Relatedly, attention also needs to be given to the quotes included in the article – at times the evidence provided doesn’t seem to be the best fit for the point being made in the narrative (for example, p.6 line 302 seems more about the importance of social support networks and less about hopelessness). Editing long quotes would help tighten the evidence presented to support arguments (see p.7 line 352). The title makes reference to life stories as a method, however there is no discussion of this approach within the methods section, and only appears later on page 18 (line 863).

The authors have now edited the methods section for clarity and to include additional information suggested by the reviewer. It has now been made clear that both studies include original data that was collected for previous dissertation work of the authors, but is now being used for the purposes of publishing this article; the sample size, demographic information about participants, recruitment and data collection details have now been included.

The title makes reference to life stories as a method and the authors believe that conducting ethnography and in depth interviews with young people is a significant method for capturing life stories. We have also gone through the quotes originally included for the article, ensuring that the quote fits alongside the heading and subheadings provided.

  1. Results

There is a lot of information included in this section and important findings seem to get lost in the detail. Restructuring will help to refine focus (see comments above). After reading the article it was still unclear to me how the experiences of homelessness and the housing need of Indigenous youth differs from that of other populations – which seems central to the argument. Later in the section there also appears to be quotes from Indigenous Elders, which was surprising given the methods seem to suggest qualitative interviews were only conducted with youth experiencing homelessness (which also seem to include people who were not Indigenous? P. 4 line 164). The data presented here seems critical to the second part of the title relating to ‘holistic healing’. It was very interesting to read about these interventions from the perspective of Elders (for example p.11 line 552). It would be useful to have more data presented here from the youth themselves on how they perceived interventions involving ‘holistic healing’ – there seems to be a short discussion here (starting p.11 line 523) but the quotes provided here all seem to revolve around negative experiences. If the article is promoting the use holistic healing practices as a more effective means to address the mental health needs of Indigenous homeless youth, then I would have expected more positive experiences being presented as evidence for that conclusion.

The authors believe that the introduction and review of literature provides a very clear discussion about how the experience of homelessness and the housing needs of Indigenous youth differs from that of other populations, due to the history of colonization and intergenerational trauma. This has been made explicitly clear in the introduction, review of literature and in the recommendations and insights from elders as this is indeed one of the central arguments of our article.

With respect to the quotes from Indigenous elders, we have included some revisions to our abstract and methods to reflect why we have included quotes from elders. The quotes from elders are specifically from Gabriel’s study. In terms of non-Indigenous youth being interviewed, Mirza has now clarified that although non-Indigenous youth were interviewed throughout her study, that only the narratives of 11 Indigenous youth from her study will be included for the purposes of this paper.

The interviews with youth and elders presented here do not include the narratives of youth in terms of how they perceived the interventions that elders spoke of. Perhaps having elders intervene with youth who are homeless and conducting that research to understand youth perspectives would be a research project the authors can consider for future research.

  1. Conclusions

Section 5 (conclusion) needs to connect specific findings (articulated within a theoretical/conceptual framework) to specific recommendations (for policy, practice, research etc). As currently presented, section 4 (discussion and recommendations) seems to include elements of the conclusion. The statement on page 15 (line 738) that ‘it can be difficult to know how best respond’ to the problem suggests that more thinking is required here to fully set out implications for policy or research agendas concerned with Indigenous homeless youth, in Canada or elsewhere.

We believe that it is in fact difficult to know how to best respond to the needs of young Indigenous people who are homeless and that more thinking is required here to fully set out implications for policy of research agendas concerned with Indigenous homeless youth, in Canada or elsewhere; to do this work is a work in progress. 

  1. Editing/language

Some edits are required for clarity for example:

We have tried our best to attend to your suggested edits in terms of language.